# Thermophilic Exopolysaccharide Films: A Potential Device for Local Antibiotic Delivery

**DOI:** 10.3390/pharmaceutics15020557

**Published:** 2023-02-07

**Authors:** Joseph M. Laubach, Rajesh K. Sani

**Affiliations:** 1Department of Biomedical Engineering, South Dakota School of Mines and Technology, Rapid City, SD 57701, USA; 2BuG ReMeDEE Consortium, South Dakota School of Mines and Technology, Rapid City, SD 57701, USA; 3Department of Chemical and Biological Engineering, South Dakota School of Mines and Technology, Rapid City, SD 57701, USA

**Keywords:** topical antibiotics, drug delivery, exopolysaccharide, polymer films, thermophile, *Geobacillus* sp., amikacin

## Abstract

Natural polysaccharides being investigated for use in the field of drug delivery commonly require the addition of sugars or pretreated biomass for fabrication. *Geobacillus* sp. strain WSUCF1 is a thermophile capable of secreting natural polymers, termed exopolysaccharides (EPSs), cultivated from cost-effective, non-treated lignocellulosic biomass carbon substrates. This preliminary investigation explores the capabilities of a 5% wt/wt amikacin-loaded film constructed from the crude EPS extracted from the strain WSUCF1. Film samples were seen to be non-cytotoxic to human keratinocytes and human skin-tissue fibroblasts, maintaining cell viability, on average, above 85% for keratinocytes over 72-h during a cell viability assay. The drug release profile of a whole film sample revealed a steady release of the antibiotic up to 12 h. The amikacin eluted by the EPS film was seen to be active against *Staphylococcus aureus*, maintaining above a 91% growth inhibition over a period of 48 h. Overall, this study demonstrates that a 5% amikacin-EPS film, grown from lignocellulosic biomass, can be a viable option for preventing or combating infections in clinical treatment.

## 1. Introduction

Bacteria of the genus *Geobacillus* are Gram-positive, spore-forming, aerobic, or facultative anaerobic, and can grow within a range of 45–75 °C [1,2,3]. The *Geobacillus* sp. strain WSUCF1 used in this study is a ligninolytic thermophile previously isolated from a compost facility at Washington State University in Pullman, WA [4]. It is recognized that microbial exopolysaccharides (EPSs) protect the cell against adverse environmental conditions, assist with adhesion, and contribute to cell-to-cell interactions [5,6]. Thermophiles generate EPSs to encapsulate their bacterial cells and form a boundary to retain hydration against the desiccation effects of thermophilic conditions [7,8]. Until now, it has been well understood that utilizing microbial EPSs in research involves bacterial cultures that require expensive carbon substrates, such as sugar, or pretreated biomass. *Geobacillus* sp. WSUCF1 is capable of consuming lignocellulosic biomass to produce EPSs without the need for pretreatment [9].

Synthetic biodegradable polymers are widely used in drug delivery and related research. The more common of these polymers include, but are not limited to, polylactic acid (PLA), polyethylene glycol (PEG), and poly(lactic-co-glycolic acid) (PLGA). The likes of PLGA, PEG, and PLA have been proposed for the fabrication of drug delivery devices, but they are not without their drawbacks [10,11,12,13]. The major concerns involve unwanted polymer–drug interactions, harmful degradation products, poor thermal properties, purification necessity, non-biodegradability, and high burst release of the drug [14,15,16,17,18,19].

EPSs have been noted to degrade within the body via natural biological processes. With that, one could deduce that the body would naturally remove the drug delivery system once the active compound has been released [20]. It has been widely reported that the research with EPSs has yielded results that revealed next to no toxicity of any kind [5,21,22,23,24,25]. Furthermore, the majority of polysaccharides are naturally antimicrobial and are easily replicable from several known, cheap, and plentiful sources [21,26,27]. When considering drug delivery technologies, film-forming abilities can entrap biological compounds and, because of its oxygen barrier properties, these entrapped molecules will remain stable and with an enhanced shelf-life if discussing industry use [20,24,28].

EPSs are a key component to the total extracellular matrix, providing the basis for the formation of biofilms. A vast increase in antibiotic resistance is seen in parallel to the formation of biofilms [29,30,31]. Biofilm-related infections are a concern especially in the clinical setting due to this reduced antibiotic susceptibility [29]. One could theorize that leftover bacteria from the EPS drug delivery device could enter any damaged tissue present on the skin’s surface, multiply to form a biofilm, enter lower layers of the skin, and damage deeper tissue or enter the bloodstream. However, this has yet to be investigated with this proposed EPS device as the idea is still in its early stages.

The overall purpose of this research is to examine the capabilities of a film comprised of thermophilic exopolysaccharides deriving from nontreated corn stover as an outline for a potential topical drug delivery tool. We employed a simple, self-forming film casting procedure utilizing a glycerol plasticizer that sharply expedites the process from reagents to a poured film and allows for minimal attention or intervention. It was hypothesized that the numerous reactive functional groups within a polysaccharide’s structure would allow for ease of chemical modification resulting in competent film forming. Furthermore, it was hypothesized that the natural antimicrobial property and biological compatibility of crude EPSs would result in noncytotoxic interactions with healthy mammalian cells. This would conclude that EPS purification would not be a necessary step before film forming. Additionally, the hydrophilicity of the EPS, although tunable, would result in the adequate release of the total drug loaded, thereby minimizing waste.

## 2. Materials and Methods

### 2.1. Materials

Amikacin was purchased from Cayman Chemical (Ann Arbor, MI, USA). Human primary epidermal keratinocytes (PCS-200-011), human fibroblasts (CCL-110), dermal basal cell medium, keratinocyte growth kit, Eagle’s minimum essential medium, and *Staphylococcus aureus* (ATCC 29213) were all purchased from ATCC (Manassas, VA, USA). Polyhydroxybutyrate (PHB) and Whatman paper filters (22 µm pore size) were purchased from Sigma Aldrich (St. Louis, MO, USA). Fetal bovine serum, Invitrogen ActinRed™ 555 ReadyProbes™ reagent, NucBlue™ Fixed Cell Stain ReadyProbes™ reagent, ProLong™ Gold antifade reagent, Image-it™ kit, CYQUANT LDH Cytotoxicity Assay Kit, and Vybrant™ MTT Cell Proliferation Assay Kit were purchased from Thermo Fisher Scientific (Waltham, MA, USA). All other chemicals and solvents were of analytical reagent grade.

### 2.2. Precipitation and Extraction of the EPSs

The *Geobacillus* sp. strain WSUCF1 media was prepared in a 1000 mL flask containing 6 g of shredded corn stover, 20 g yeast extract, 3 g NaCl, and 1000 mL of distilled (DI) water [9]. The corn stover, yeast extract, and NaCl solution was first made homogeneous using a magnetic stir bar with the pH set to 7.0, then allowed to autoclave at 121 °C for 20 min. Once the media had cooled, the flask was inoculated with the WSUCF1 bacteria and placed in a shaker set at 60 °C for 24 h. After 24 h, the liquid culture was passed through a Büchner funnel with the bottom of the funnel lined with a 0.22 µm pore size Whatman paper filter to separate the corn stover from the culture.

The bacterial cells were separated from the biofilm matrix-containing supernatant, which includes the EPS, via centrifugation at 8000× *g* for 20 min in 50 mL tubes or sometimes 500 mL centrifuge bottles. To separate the media from the EPS further, an optional step involves allowing the supernatant to evaporate via rotary evaporator until approximately 250–300 mL of water was removed. To precipitate the crude EPS, a 1:1 ratio of absolute ethanol was added to the remaining supernatant and allowed to sit in a freezer overnight at −20 °C. Lastly, the crude EPS was obtained by centrifuging the ethanol/supernatant solution at 8000× *g* for 40 min [9]. Following collection, the crude EPS was stored at −20 °C without a cryoprotectant.

### 2.3. Casting of the Amikacin-Loaded Film

The crude EPS was added to a 50 mL centrifuge tube and mixed with distilled water to achieve 5% (*w/v*) concentration. Glycerol was added as a plasticizer at a concentration 30% (g glycerol/g crude EPS). Forty percent *w/w* (g EDC/g crude EPS) 1-ethyl-3-(3-dimethylaminopropyl)carbodiimide hydrochloride (EDC) was added to the solution as a crosslinker (100 mM concentration), followed by 5% *w/w* amikacin (mass amikacin/total mass of solution). A total of 20 mL of the homogenous EPS film solution was poured into a 3.1-inch diameter PTFE evaporating dish and allowed to cure within a dry heat oven for 36 h. EDC is a known crosslinker that has previously been reported to have minimal toxicity and has been noted to modify side-chains of proteins and polysaccharides, specifically those containing a carbonyl group [32]. Moreover, the addition of carbodiimides has been studied extensively for the manufacturing of biomaterials [33,34].

### 2.4. Cell Culture

PCS-200-011 human keratinocytes and CCL-110 human fibroblasts were purchased from American Type Culture Collection (ATCC, Manassas, VA, USA). PCS-200-011 cells were maintained in dermal bell basal medium (PCS-200-030) supplemented with a keratinocyte growth kit (PCS-200-040), fetal bovine serum (10%), penicillin (100 units/mL), and streptomycin (50 units/mL) at 37 °C in a humidified (5% CO_2_, 95% air) atmosphere. CCL-110 cells were maintained in Eagle’s minimum essential medium supplemented with fetal bovine serum (10%), penicillin (100 units/mL), and streptomycin (50 units/mL) at 37 °C in a humidified (5% CO_2_, 95% air) atmosphere. The cell lines were cultured and maintained according to the specifications of ATCC.

### 2.5. Fourier-Transform Infrared Spectroscopy

The Fourier-transform infrared spectroscopy (FTIR) spectra of a standard EPS film, EPS + EDC film, and EPS + EDC + amikacin film were recorded using a Nicolet iS10 (Thermofisher, Waltham, MA, USA) FTIR spectrometer. The resolution of this instrument is 0.4 cm^−1^. Sterile tweezers were used to remove a small sample from the center of the film for analysis. A total of 16 sample scans were performed from 4000 to 499 cm^−1^.

### 2.6. MTT Assay

An Invitrogen Vybrant™ MTT cell proliferation assay kit was used to perform an MTT assay on the cell lines exposed to the varying film types. The protocol was performed as per manufacturer’s guidelines. Assay data were collected using an Epoch 2 microplate spectrophotometer (Biotek, Winooski, VT, USA). A keratinocyte cell suspension of 0.23 × 10^6^ cells/mL was prepared in fresh media. A fibroblast cell suspension of 0.2 × 10^6^ cells/mL was also prepared. Into a 24-well plate, 500 µL of either cell suspension was added. The cells were then given an incubation time of 60 h. Positive controls (100% viability) were examined without the presence of the film. Negative controls had 50 µL of 100% hydrogen peroxide added. Both cell lines were exposed to both the EPS film-forming solution as well as an amikacin solution in DI water of equal concentration to the film. The experiment was performed in quadruplicate (*n* = 4).

For the exposure portion of the experiment, 50 µL of the complete film-forming solution was added to the wells. The cells were not examined against sections of a cured film due to previous experiments within our lab where the film dissolved within the media over time, leaving remnants of itself behind within the well, even after a media change. This had the potential to affect the ability of the MTT assay as well as the eventual analysis with a microplate reader. Physical film pieces, all measuring 0.7 in × 0.4 in and approximately 50 mg, were used for PHB film exposure experiments. The PHB films did not dissolve during incubation with keratinocytes, and subsequently were easily removed and discarded from the wells using sterile tweezers once the respective time point had been reached.

### 2.7. LDH Assay

With the intent of providing further evidence for the lack of cytotoxicity, a lactate dehydrogenase (LDH) assay was performed, as per the manufacturer’s guidelines. The assay was performed using the same cell lines as the MTT assay from Section 2.5, but in a 96-well plate instead of a 24-well plate. Assay data were again collected using an Epoch 2 microplate spectrophotometer (Biotek, Winooski, VT, USA).

A cell suspension of 6000 cells per 100 µL of fresh media was added to each well. The seeded cells were allowed 48 h of incubation. Chemical treatment consisted of 10 µL of either the film-forming solution or an amikacin solution in DI water equal to the concentration of the film solution, as was the protocol during the MTT assay. The experiment was performed in triplicate (*n* = 3). The amount of lactate dehydrogenase present in the medium was determined by measuring the absorbance at 2 wavelengths (490 nm and 680 nm) using a microplate reader.

### 2.8. Fixed-Cell Fluorescence Imaging

Round (22 mm) collagen-coated coverslips (BioCoat, Bedford, MA, USA) were placed into the wells of a 6-well plate for cells to adhere to. Cells were cultured at a seeding density of 0.3 × 10^6^ to a total volume of 3 mL in each well, media included. Cells were allowed 48 h to reach confluency. Controls did not contain EPS samples. For the exposure experiments, 100 µL of crude EPS in molecular biology-grade water (5% *w/v*) was added to the well.

At 12 and 36 h, cells were fixed before either actin or nuclear epifluorescent stain was applied. For controls, cells were stained, but were not subject to EPS exposure. All images were not doctored in any way. All photographs of F-actin were taken using the same microscope camera settings. All photographs of DAPI staining were taken using the same microscope camera settings. Images were acquired using an Olympus IX50 inverted fluorescence microscope (Olympus, Japan). ActinRed™ 555 ReadyProbes™ reagent, NucBlue™ Fixed Cell Stain ReadyProbes™ reagent, and Image-it™ kit (Life Technologies Corp., Eugene, OR, USA) were used to examine the cellular structure of the keratinocytes and fibroblasts when exposed to the amikacin-loaded film.

### 2.9. HPLC Drug Release Profile

The rate of amikacin release from the film was determined by submerging a 3.1-inch diameter, newly casted sample of amikacin-loaded EPS film in 50 mL of PBS (1 × 0.01 M, pH 7.4) within a 50 mL centrifuge tube. The tube was incubated at 37 °C and agitated at 100 RPM. Following this, 1 mL samples were extracted from the centrifuge tube at specific time points and stored in a refrigerator freezer until all time point samples were obtained. Prior to HPLC examination, all samples were sterile filtered.

The mobile phase consisted of 47% acetonitrile, 53% HPLC grade water, and 1 mL/L acetic acid. The pump was set to 0.6 mL/min, the detector set to 365 nm, and the column temperature was set to 40 °C. A 300 mm length × 7.8 mm inner diameter Aminex HPX-87H column was used within a Shimadzu Prominence-i LC-2030c plus liquid chromatograph (Shimadzu, Kyoto, Japan).

In triplicate, 1 mL samples of the solution in the tube were removed and added to a HPLC vial. The vials were placed in a refrigerator freezer for storage until the experiment was complete and the samples were ready to be analyzed by HPLC. To calculate the concentration of amikacin found in the samples, amikacin standards were read using the same HPLC method, the area of their respective peaks were graphed, and the equation of the trendline was used to ultimately calculate the amikacin found in each unknown.

### 2.10. Staphylococcus aureus Turbidity Assay

A turbidity assay was performed similar to that of Smith et al. [35]. In triplicate, *S. aureus* was exposed to the elution, a high concentration of amikacin in water, and without the presence of amikacin. Using the elutions from the HPLC study, 200 µL of elution containing the antibiotic was added to a conical tube already containing 1.8 mL of sterile tryptic soy broth. Each tube was then inoculated with 20 µL of *Staphylococcus aureus* (ATCC 29213). All tubes were placed in an incubator set to 37 °C until their respective time points were reached. Once removed from the incubator, the conical tubes were vortexed and a 1 mL sample from the tube was added to a cuvette.

A Spectronic 200 spectrophotometer (Thermofisher, Waltham, Massachusetts) was used to measure the OD600. Controls and values were normalized by the optical density at 600 nm (OD600) of an initial sample of *S. aureus* allowed to grow for 24 h prior to the experiment taking place. The absorbance of *S. aureus* at OD600 after 24 h of growth was measured to be 1.360. For the blank, a sample containing only tryptic soy broth was scanned. Negative controls were represented as *S. aureus* growth exposed to 200 µL of amikacin at a high concentration (1.70 g amikacin in 10 mL DI water). Positive controls were represented as *S. aureus* growth left uninterrupted and unexposed to any additional applications.

### 2.11. Quantity of EPS Produced from Corn Stover vs. Dextrose

Over the course of this project, the EPS produced per liter of media was recorded when utilizing either corn stover or dextrose as the primary carbon source. Traditionally, the primary limitation of utilizing microbial EPSs involves the bioprocessing methods of EPS where expensive carbon-rich feedstocks, such as sugars, are required. *Geobacillus* sp. WSUCF1 is capable of consuming lignocellulosic biomass, such as corn stover, to produce EPSs without the need for pretreatment.

### 2.12. EPS/Amikacin Film Thickness

A digital vernier caliper was used to measure the thickness of the films produced during this study. Each film was removed from the PTFE evaporating dish and the vernier caliper was placed in the center of the film to measure thickness. The vernier caliper used is accurate to 0.02 mm, measuring 0–150 mm.

### 2.13. Swelling Ratio of the Film

To measure swelling of the EPS/amikacin films, the weights of whole films were measured in the presence and absence of amikacin. Five films without amikacin and five films with amikacin in the formulation were developed and weighed. The films were weighed in their respective PTFE evaporating dishes after being considered fully cured. The weight of the empty PTFE dish was subtracted from the weights of the films in the dishes to obtain accurate values for the film’s mass. A well-crosslinked polymeric device should display minimal swelling.
Swelling ratio (%) = (final film weight − initial film weight)/(initial film weight) × 100

### 2.14. Statistical Analysis

Experiments were performed in triplicate or quadruplicate, and statistically significant differences (*p* < 0.05) were determined using 1-way or 2-way repeated measures ANOVA followed by a Tukey’s post hoc test. Data were analyzed using OriginPro 2022 software.

## 3. Results and Discussion

### 3.1. Features of Various EPS Films

The addition of the EDC crosslinker noticeably altered the physical characteristics of the film, including increased flexibility (Figure 1A–C). In Figure 1A, the EPS film prior to the introduction of the EDC crosslinker is noticeably frail and brittle. Figure 1 illustrates the addition of EDC without amikacin. As shown, the film’s structure is remarkably elastic. Figure 1C is the product of the addition of amikacin and EDC. Once again, a change in texture can be seen, undoubtedly due to the EDC’s interaction with amikacin.

Amikacin is an aminoglycoside with broad treating capabilities that are effective against Gram-negative bacterial infections. One such bacteria amikacin is effective against *Staphylococcus aureus*, in which thousands of infections are reported each year by American hospitals [36]. Aminoglycosides are known to be susceptible to cleavage of their glycoside bonds via oxidation [37,38]. The oxidative properties of hydroxyl radicals are increased within an acidic solution [39]. As is mentioned in the introduction, these extracellular substances exhibit a type of oxygen barrier property that theoretically could protect the amikacin molecules from oxidation once the film is formed. However, atmospheric oxidation can still potentially occur during the process of mixing together the film’s reagents and is something that should still be considered. The amikacin-loaded EPS films were thin, flexible, and could easily be removed from the PTFE dish by using a pair of simple tweezers. Recreated films also exhibited identical features, confirming consistent reproducibility of the manufacturing process. It makes sense that films and patches intended for topical and transdermal antibiotic delivery should be flexible enough to fit around the skin, but also tough enough to not be compromised with the movement of the skin [40,41].

### 3.2. Uptake of EDC and Amikacin by FTIR

Figure 2 displays the IR spectra from the examination of the EPS film, EPS and EDC film, and EPS, EDC, and amikacin film. The values for the major peaks of the EPS amikacin-loaded film can be seen. The spectra from the amikacin-loaded film shows a peak at 1702 cm^−1^. This peak indicates a strong carbonyl stretching [42]. The structure of amikacin does indeed contain a carbonyl functional group. The structure of EDC does not. To explain the low transmittance of the carbonyl peak, the particular film used in this study contained a 0.5% amikacin concentration rather than a 5% concentration that was used in the remainder of the results. This is simply because this was the concentration that the initial films were made with.

The second significant peak lies at 2126 cm^−1^. Peaks in this region often represent carbon interacting with nitrogen [42]. Both the structures of EDC and amikacin contain potential azide, nitrile, or carbodiimmide functional groups, including during their respective intermediate states while interacting with fellow compounds within the mixture. EDC is a carbodiimide crosslinking compound which undergoes carbodiimide conjugation during a reaction with a carboxylate or primary amine group [43,44].

The broad spectrums seen in all spectra between 3276 and approximately 3300 cm^−1^ indicate stretching of an O–H bond. The dual peaks at 2934 cm^−1^ specifies the C–H stretching within the pyranose ring of polysaccharides. The immense peaks from all three spectra that can be seen around 1026 cm^−1^ are the result of glycosidic bonding between monosaccharide units that make up polysaccharides [9].

The peaks at 1251, 1558, and 1632 cm^−1^ illustrate specific amide stretching for carbonyl bonds [9,42,45]. The structure of EDC contains a primary amine that can form new amide bonds during a reaction with another compound, in this case, the EPS. The peak at 1462 cm^−1^ provides evidence for C–H bending primarily seen in alkane compounds [42]. The spectra in Figure 2 of both the EDC film and the EDC + amikacin film differ from the EPS film in that several new peaks exist. This also provides evidence of uptake and reaction among the EPS, EDC, and antibiotic. The shrinking of peaks representing functional groups of the EPS/amikacin film in comparison to the EPS/EDC-only film is a sign of successful and effective crosslinking.

### 3.3. MTT Assay

To determine the metabolic activity of human keratinocytes and human skin tissue fibroblasts when exposed to the EPS film, an MTT colorimetric assay was performed. MTT assays measure the cellular metabolic activity, but will be reported as cellular viability percentage, as is common. The general aim of this study was to examine the possibility of an EPS-generated film as a tool for topical, and eventually transdermal, drug delivery. This film would immediately encounter the skin upon use.

Keratinocytes were selected due to the fact they are the cells that make up the epidermis. Dermal fibroblasts are a type of epithelial cell that play an important role to generate connective tissue and assist in cutaneous wound healing within the dermis [46]. As a parallel experiment to compare the results of exposing the cells to a solvent-casted film, vs. a glycerol plasticizer, a polyhydroxybutyrate (PHB) film was constructed using a method by Anbukarasu et al. [47].

The results of these experiments are shown as Figure 3a,b. Both cell lines showed a high viability percentage throughout the experiment when exposed to the EPS film, with a minor decrease in viability being recorded for both cell lines at the 48 and 72-h mark for those treated with the antibiotic solution. Regardless, this decrease still resulted in a viability of above 80%, as an average for both the keratinocytes and fibroblasts. Interestingly, experiments for both cell types at several time points yielded higher viability when exposed to the amikacin-loaded EPS film solution vs. the antibiotic solution. This may point towards the crude EPS offering protection or supplementation to the cells.

As was mentioned earlier, previous research on the cytotoxic nature of EPSs had resulted in near-zero values. These results are important to the idea of EPS-related research carrying over into the fields of biomaterial and drug delivery devices. As both keratinocytes and fibroblasts are important contributors to wound repair on the skin, it is beneficial to know that a potential EPS–amikacin patch will not harm these two cell lines to a certain degree and should not inhibit wound healing. Overall, there is not much variability between the amikacin-loaded EPS film and the amikacin solution, but again, the goal of this experiment was to provide evidence that EPS will not harm the studied keratinocytes and fibroblasts after conjugation with EDC and amikacin.

Appendix A details the results of keratinocytes exposed to an EPS film without the inclusion of amikacin. A control film containing all the typical film reagents, without amikacin, was added to keratinocytes under the same experimental parameters. For the duration of the experiment, the results saw the cell line maintaining nearly 90% viability on average. As EDC is a known crosslinking reagent that is minimally cytotoxic, these results align with the previously reported data [32,45,48,49]. Moreover, Wang and company previously concluded a relative lack of cytotoxicity in all individual EPSs that make up the crude EPS from WSUCF1 [9].

The keratinocytes did not fare as well when exposed to the plain PHB film. The results are displayed in Figure 3c. Viability immediately decreased and never averaged close to the values of the EPS film exposure over the duration of the experiment. It is possible that this is the result of residual acetic acid within the framework of the film that is required for casting. It can be believed that the same results would be seen in other epithelial cells. To possibly explain the individual viability variation for each time point, it is believed that the potential residual acetic acid may have been unevenly distributed within the cured film that was used for this assay. Pieces of the PHB film were excised from arbitrary locations of the whole film, contributing to the unpredictability of data. The use of a noncytotoxic plasticizer, such as glycerol, appears to be far more beneficial and less harmful than the utilization of a volatile organic compound, such as acetic acid. Although PHB is of microbial origin, much like the thermophilic EPS, PHB is water insoluble and requires a solvent in the form of organic compounds, alcohols, or weak acids. Although this is only one of many types of biopolymers, the important takeaway is the minimal or zero cytotoxicity that the EPS has shown in the results of the MTT assay.

### 3.4. LDH Assay

The results of the LDH assay are shown in Figure 4a,b. The outcomes of the assay provided further evidence that the EPS/amikacin films displayed minimal cytotoxicity relative to the values provided by the amikacin in water solution. Moreover, the results displayed here mirror the results of the MTT assays with regards to similar cell viability/cytotoxicity values calculated for the EPS/amikacin film. As an average, the cytotoxicity value for the experimented keratinocytes peaked at the 48-h time point with a value of 12.26% with regards to film exposure. During the assay with the fibroblasts, cytotoxicity peaked at the 72-h time point with a value of 12%, also calculated by averaging the three values at this time. This assay provides further evidence that this current draft of a drug-loaded EPS film should not harm healthy keratinocytes and dermal fibroblasts during the body’s process of skin tissue repair. More so, this device would not be the cause for skin damage when applied to a patient given the current data. As was mentioned, potential biofilm formation could result in serious infections. Although these in vitro findings cannot provide direct evidence on whether or not the patient will be at high risk of unwanted biofilm formation, these data still provide a foundation that encourages the continuation of EPS-derived drug delivery devices.

### 3.5. Fixed-Cell Fluorescence Imaging

To rule out if this specific thermophilic EPS could have an effect on skin cell morphology and cellular structure, keratinocytes were exposed to 100 µL samples of crude EPS dissolved in molecular biology-grade water (5% *w/v*) for periods of 12 and 36 h. The addition of undissolved crude EPS was not used after previous experiments experienced clouded and otherwise compromised coverslips after the crude EPS had somewhat dissociated within the cell media. Complete EPS/amikacin film samples were not studied here because the results could have been due to any of the film-forming solution’s components.

Figure 5B,C represents the acquired data for cell fluorescence intensity for 12 h and 36 h, respectively, after human keratinocytes were exposed to an EPS solution equal to that used in the construction of the amikacin films (5% *w/v*). Data are expressed as individual data points plus the mean. It can be seen in the chart that there appears to be duplicates of each dataset. Each coverslip had two locations on the coverslip examined via the microscope. Within these images, seven random individual cells from seven different quadrants (top left, top center, top right, center, bottom left, bottom center, bottom right) of the picture had their fluorescence intensity measured using ImageJ.

Once the data from the ImageJ results window had been transferred to a Microsoft Excel document, the following formula was used to determine corrected total cell fluorescence (CTCF):CTCF = integrated density − (area of selected cell × mean fluorescence of background readings)

As a parallel experiment, four new random locations on each coverslip were selected, and from those locations, the cells that were in view were counted using the DAPI stain. The purpose here was to note any change in cell density or loss of cells between control and exposure groups. The results are documented in Table 1. As can be seen in the table, there does exist a slight loss of keratinocytes within the exposure group when comparing to the values of the control (untreated) group. These data reflect what was seen in the cell viability and cytotoxicity assays where the data reported a minimal, but present, reduction in healthy cells.

To narrow down the cause of potential cellular harm to strictly the EPSs, this experiment was performed by exposing the cells to only crude EPSs that had been dissolved in water. Performing this in vitro assay using portions of a fully cured and complete film could potentially lead to cellular deformation being caused by other components of the film. Furthermore, it was found that performing this experiment with undissolved crude EPS and film samples resulted in residue being found on the coverslips. This experiment relies heavily on the fluorescence intensity of color, so we believed it would be more accurate to pre-dissolve the EPS before exposure, which is undertaken initially during the creation of the film-forming solution.

At the 12-h mark, the values for F-actin and DAPI fluorescence not exposed to EPS appear to double, on average. However, as can be seen in Figure 5A, the keratinocytes look to be confluent and healthy in appearance in both scenarios. The individual filaments of the cells that had not been exposed to EPS appear to be more defined. This could be attributed to EPS residue that attached onto the coverslip during the exposure period.

Actin and DAPI fluorescence values for keratinocytes both exposed and not exposed to EPS were seen to be less, on average, than their 12-h counterparts. Interestingly, the actin and DAPI fluorescence data for both exposure groups were recorded as being more similar to each other after 36 h than what was seen after 12 h. Regardless, the cells appear to be highly confluent with normal features, as seen in Figure 5A. A common source of error with fluorescence studies involves photobleaching and subsequent degradation of dyes. One could point to photobleaching or poor camera contrast when explaining a potential reason behind the decrease in actin and DAPI fluorescence intensity.

The proposed purpose of this EPS drug delivery device would bring it into immediate contact with the human skin. Subsequently, it was important to understand the result of in vitro contact between the polysaccharide and human skin cells. The MTT and LDH assays did provide evidence for the film having a relatively noncytotoxic characteristic towards primary skin cells, but we believed that did not determine whether exposure to the EPSs would disrupt cell integrity. Actin is known to be the most abundant protein in eukaryotic cells. It is said to play a part in a plethora of important roles spanning from the preservation of cell shape, to maintaining cell polarity, to regulating transcription [50]. For the proposal of EPSs as the primary building block for biomedical devices, it is encouraging for the preliminary data to demonstrate, at worst, a minimal negative effect on keratinocyte phenotype. Here, with fluorescence staining, we can both visualize the potential change in the structure of the cells as well as count the cells present using DAPI staining. The data can then be compared between treatments and controls. Furthermore, these data support the data of the MTT and LDH assays in a parallel manner.

### 3.6. HPLC Drug Release Profile

The total amikacin release peaks at approximately 93% at 12 h. The cumulative results are shown in Figure 6. This value is an average of three recorded elution samples. Noel et al. (2008) [36] reported a 5% amikacin film comprised of chitosan showed near complete release of the antibiotic at 72 h. However, the chitosan film was not constructed using a crosslinker, and the amikacin was loaded by submerging the film into an amikacin solution. Additionally, the authors did not agitate the conical tube during their drug release study as we did [36].

It can be seen that one data point exists above the maximum amikacin quantity. This could potentially be due to error resulting from buildup in the HPLC column during the reporting of the standards or unknowns up to that point. Regardless, from simply determining the point of near-complete antibiotic release, it can be agreed that knowing that exists at approximately 12 h is valuable information for determining a role in therapeutics. A gradual extended release is seen up until the 12-h time point. Extended release in drug delivery is a valuable trait. It remains to be seen if an alteration in crosslinker concentration would either extend or decrease this release rate. Furthermore, between the 12 and 24-h marks, the concentration of amikacin found in the unknown samples plummets immensely. While this concentration of amikacin could be a potential instrumental error, much like what was mentioned earlier when discussing PLA, this decrease in registered amikacin could be the result of polymer–drug interactions affecting the structure of amikacin, resulting in degradation.

Preliminary investigations suggest our EPS delivery system offers a moderate extended release rate. Additionally, the ease of film administration offers the ability to be removed at any point. A chondroitin-dextran hydrogel encapsulating amikacin (1.43 wt%) showed that acidity could play an important role in drug release mechanics. Although an extended release was maintained, the authors noted that 30% of the total amikacin loaded into the hydrogel had been released after 24 h at a pH of 5.0 [48]. In a separate study, an alginate-based hydrogel released 80% of its amikacin after 24 h while studied at a pH of 5.0 [49]. Further investigations are needed to determine the ability to modify the release dosage. As we have mentioned the structural compatibility EDC theoretically has with amikacin, it could be believed that a differing crosslinking agent would play an important role in determining rate of antibiotic release. The ratio of crosslinker used in this study was arbitrary, thus the same could possibly be said for varying quantities of EDC used here added to the film-forming solution.

### 3.7. Staphylococcus aureus Turbidity Assay

Given that there remained a large volume of elution from the EPS film after examination of the film’s release rate, the antibiotic’s effectiveness after uptake and release was determined by exposing it to samples of *S. aureus* during growth and measuring inhibition. Additionally, the drug release profile of the film estimated the point of peak drug release to be at approximately 12 h. Thus, a new film’s elution obtained after 12 h of dissolving in PBS was also used in this study as a comparison.

The results are seen in Figure 7. The elution from the EPS film is compared with the results from the negative control, a high concentration of amikacin in water (1.70 g in 10 mL DI water). The elution, even at a 5% concentration, is seen to be effective against the *S. aureus*. Furthermore, the elution from the suspected point of optimal release was seen to be more effective than the leftover elution from the HPLC studies. At the peak growth period during this study at around the three hour time point, the *S. aureus* growth maximizes at approximately 8.07% average. Another way to report this value is a 91.93% ± 0.31 inhibition of the growth. For the 12-h release elution, the maximum value allowed for *S. aureus* growth was seen at the 7-h point and reached an average of 5.88%, or 94.12% ± 0.0016 inhibition of growth. In comparison to Noel et al. (2008), their reported chitosan film elution yielded a maximum value for *S. aureus* inhibition of 85.13 ± 3.00 at the one hour time point [36]. The results of this experiment provide evidence that amikacin effectiveness is unhindered by uptake into the film’s composition and subsequent release. It can be believed that an increase in amikacin concentration during the film’s casting procedure could result in a larger inhibition percentage of *S. aureus* growth.

The turbidity assay is a bacteriostatic assay that can only quantify the bacterial growth inhibition of an elution. New experiments involving bactericidal assays would provide evidence for the film’s capability to eliminate bacteria stemming from a wound site. Furthermore, the bacteria were not supplemented with additional reagents, such as calcium or iron, which would replicate a mock environment of human skin or blood. This could have possibly altered the outcome of the experiment.

Table 2 displays the numerical values associated with this experiment as the average for each time point. It can be seen with more precision that the amikacin at complete release (12 h) is more effective against *S. aureus* growth across all time points in comparison with the 48+ h elution, albeit marginally. In a clinical scenario, if the film can administer the antibiotic effectively in a similar manner as was seen in these experiments, it appears as though this dosage of amikacin is nearly ideal for irradicating *S. aureus* infections. However, the full delivery capabilities of the film may not be fully understood until tests can be performed on skin tissue or skin substitutes.

### 3.8. EPS Produced per Liter Using Varying Carbon Sources: Dextrose vs. Corn Stover

Throughout this study, bacterial cultures were prepared in one liter intervals to obtain crude EPS. For several of the cultures that were prepared, the quantity of crude EPSs they produced was recorded by weighing the amount of EPSs present after the final centrifuging step. As a parallel experiment, thermophilic cultures were also prepared using the same concentration of dextrose as the primary carbon source, and their EPS production was recorded. The results as an average can be seen in Table 3. From the cultures examined, corn stover was seen to produce nearly 2.5 times the weight of crude EPSs as dextrose, on average. The greater “quantity” of EPSs produced by corn stover does not necessarily equate to a greater number of biopolymers. It is possible that the utilization of corn stover produces biopolymers that exhibit a greater density in comparison to those produced when using sugar. Regardless, as the films presented here are made on the basis of weight of EPSs, the use of corn stover appears to be far more effective when considering the idea of mass-producing devices and materials that may utilize this thermophilic biopolymer.

### 3.9. Film Thickness

Thickness of the films was measured at 0.37, 0.38, 0.40, and 0.41 mm (*n* = 4), for an average value of 0.39 mm ± 0.0158. The low value of standard deviation provides evidence for the consistency and uniformity of the films as they are produced. As mentioned previously, one measurement was taken in the center of each film. The film thickness was measured immediately upon removal from the dry-heat oven. The presence of air bubbles along the film structure could influence the values obtained for thickness. Although air bubble patches were avoided when measuring, their presence could have disrupted the overall thickness consistency across the film. It would be beneficial for a therapeutic EPS film to be as thin as possible without yielding negative mechanical properties. A thin film would allow for ease of concealing the film underneath clothing. As it stands now, the resulting films from this preliminary method do exhibit minimal thickness. It can be believed that by utilizing an evaporating dish with a larger diameter, or perhaps a different mold altogether, the current film-forming method could yield thinner films with possibly a more acrylic appearance.

### 3.10. Swelling Ratio

The weights obtained by the formulated films can be seen in Table 4. The drug-loaded films were noted to weigh between 1.219 g and 1.429 g. The variation in weights among the five examined films could be attributed to incomplete mixing, excess water retention, or digital scale error. The weights of the five films without amikacin were averaged to be 0.9456 g. The swelling capacity of the films ranged between 28.91 and 37.69%, with a potential outlier recorded as 51.12%.

The value for swelling ratio depicts how well these films are crosslinked. The lower the value for swelling following addition of drug and crosslinker, the more efficient the crosslinking within the device. In a separate study, chitosan/aminoglycoside films were formed using acid solvents and without the presence of a crosslinking agent. Chitosan films with 80% degrees of deacetylation, created using a lactic acid solvent, and in situ loaded with daptomycin experienced a swelling value of nearly 300%. Variations of these chitosan films using an acetic acid solvent and loaded with vancomycin equally experienced a nearly 300% swelling ratio [35]. Now that the swelling ratio has been recorded for these particular films, the next logical step would be to alter the crosslinker concentration and the crosslinker that is used altogether.

## 4. Conclusions

This study presents a novel approach for synthesizing an amikacin-loaded EPS film using exopolysaccharides extracted from the thermophile *Geobacillus* sp. WSUCF1. This approach can alleviate disadvantages involving toxicity, inadequate thermal properties, hydrophobic behaviors, and difficulties with drug conjugations that can be seen in synthetic polymers that are currently present in the field of drug delivery. Moreover, the thermophile used in this study can utilize lignocellulosic biomass as a primary carbon source during culturing, without the need for pretreatment. This film yielded near-zero cytotoxicity in tested human keratinocytes and dermal fibroblasts. The film’s elution remained effective after release against *S. aureus*, holding its growth on average below 8%, or a 92% inhibition. Continuing studies could incorporate methods to bypass the skin, possibly with the addition of a chemical permeation enhancer in the film-forming solution. Furthermore, the mechanical and physiochemical properties of the film need to be investigated. Overall, this preliminary outline provides a solid foundation with a strong allure to be built upon going forward.

## Figures and Tables

**Figure 1 pharmaceutics-15-00557-f001:**
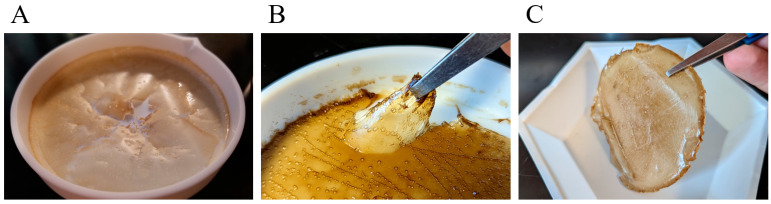
Various EPS films (**A**) EPS film without crosslinker and antibiotic. (**B**) EPS film with the addition of EDC. (**C**) EPS film containing EDC crosslinker and amikacin antibiotic. The amikacin film displays a malleable, uniform, and robust texture.

**Figure 2 pharmaceutics-15-00557-f002:**
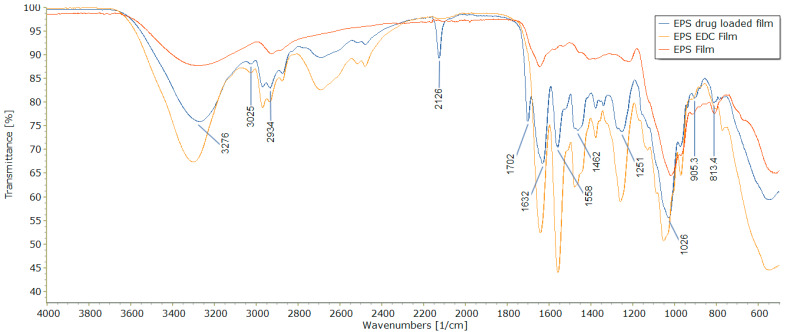
FTIR spectra of all three film types. Peaks at 1702 wavenumbers and 2126 wavenumbers indicate successful interaction of EDC crosslinker as well as uptake of the amikacin antibiotic.

**Figure 3 pharmaceutics-15-00557-f003:**
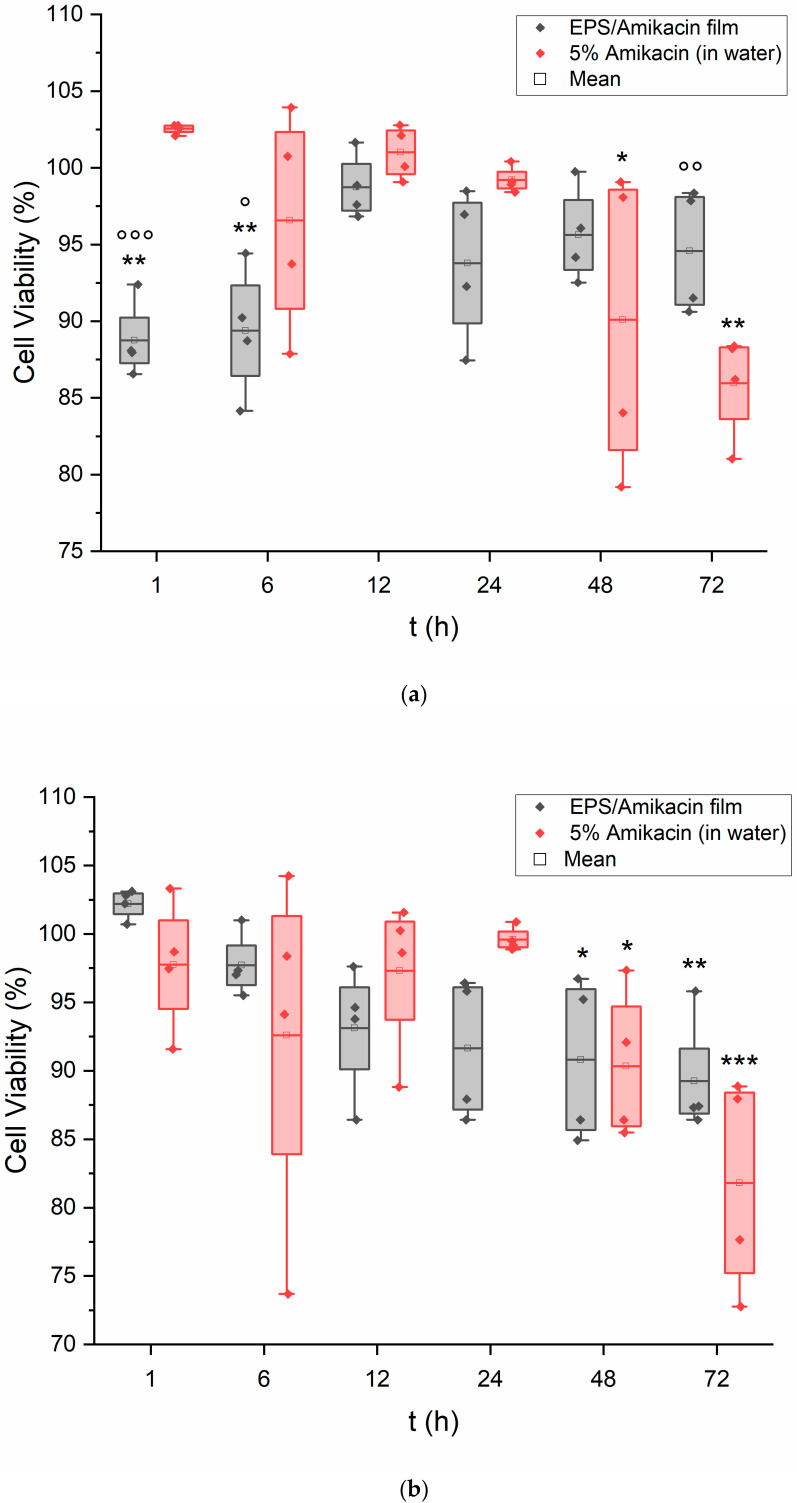
(**a**) Viability of keratinocytes exposed to amikacin. Keratinocytes exposed to the amikacin-loaded EPS film as well as an amikacin solution of equal concentration in DI water (*n* = 4). Data were also evaluated for significance using a two-way repeated measures ANOVA and Tukey’s test post hoc. Versus untreated cells: * *p* < 0.01, ** *p* < 0.001. Versus free amikacin in water: ° *p* < 0.05, °° *p* < 0.01, °°° *p* < 0.001. (**b**) Viability of fibroblasts exposed to the amikacin-loaded EPS film as well as an amikacin solution of equal concentration in DI water (*n* = 4). Data were also evaluated for significance using a two-way repeated measures ANOVA and Tukey’s test post hoc. Versus untreated cells: * *p* < 0.05, ** *p* < 0.01, *** *p* < 0.001. (**c**) MTT assay results. Keratinocytes exposed to a sample of PHB film (*n* = 4). Data were also evaluated for significance using a two-way repeated measures ANOVA and Tukey’s test post hoc. Versus untreated cells: * *p* < 0.001, ** *p* < 0.00001.

**Figure 4 pharmaceutics-15-00557-f004:**
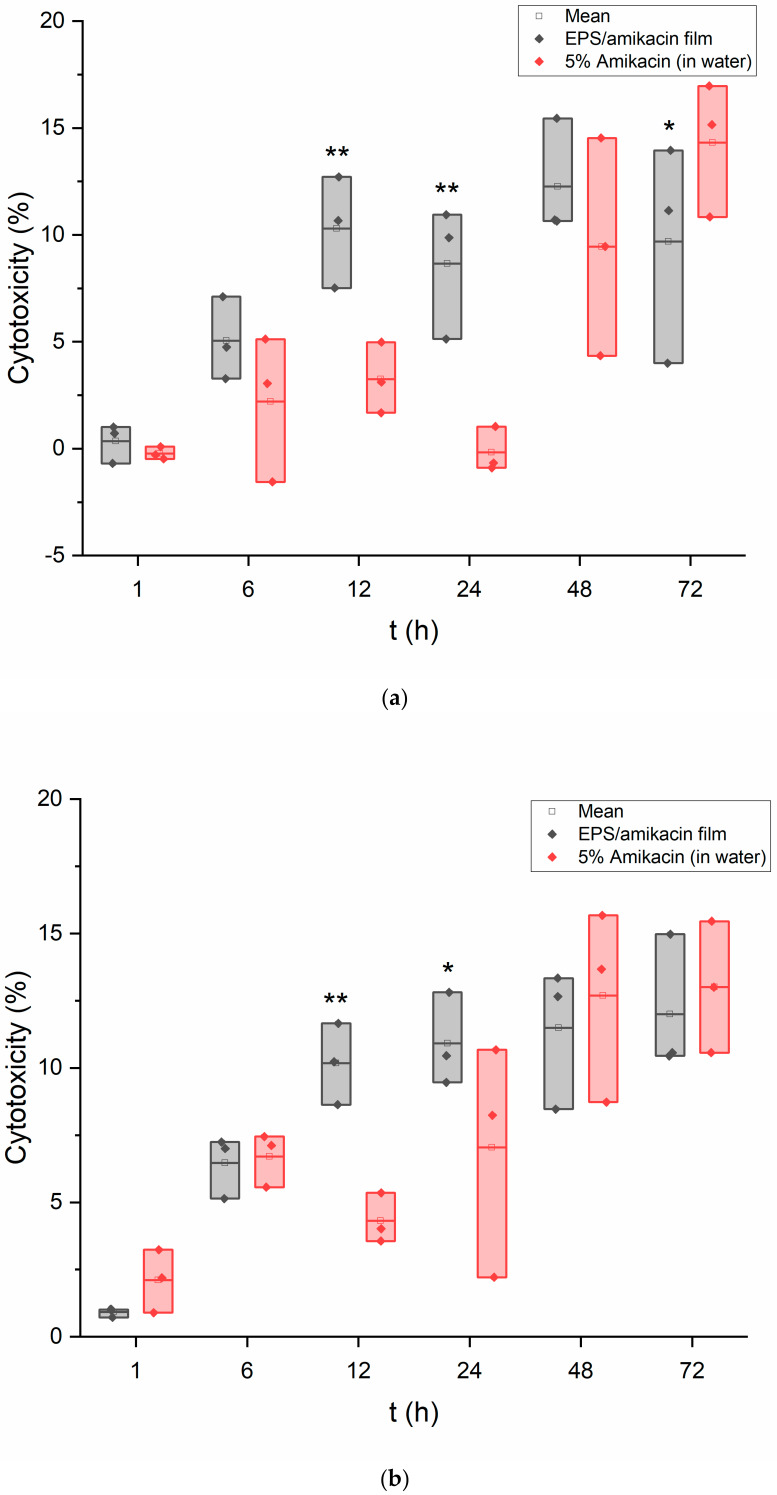
(**a**) LDH assay results of keratinocytes exposed to EPS/amikacin film-forming solution as well as a 5% amikacin solution in water (*n* = 3). Data were also evaluated for significance using a two-way repeated measures ANOVA and Tukey’s test post hoc. Versus untreated cells: *p* < 0.001. Versus free amikacin in water: * *p* < 0.05, ** *p* < 0.001. (**b**) LDH results of fibroblasts exposed to EPS/amikacin film-forming solution as well as 5% amikacin solution in water (*n* = 3). Data were also evaluated for significance using a two-way repeated measures ANOVA and Tukey’s test post hoc. Versus untreated cells: *p* < 0.001. Versus free amikacin in water: * *p* < 0.05, ** *p* < 0.01.

**Figure 5 pharmaceutics-15-00557-f005:**
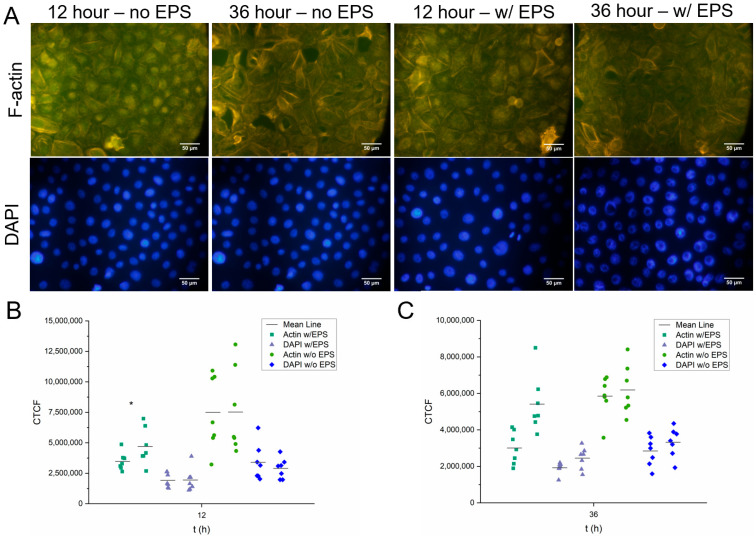
Results of fixed-cell fluorescence staining. (**A**) F-actin and DAPI degradation was evaluated on human keratinocytes after exposure to an EPS solution (5% *w/v*). (**B**) Corrected total cell fluorescence after 12 h with and without EPS exposure (*n* = 7). (**C**) Corrected total cell fluorescence after 36 h with and without EPS exposure (*n* = 7). Individual cells were chosen at random to obtain data. Quantification was undertaken with the help of ImageJ software. Data for both time points were also evaluated for significance using a one-way repeated measures ANOVA and Tukey’s test post hoc (actin and DAPI vs. respective controls: *p* < 0.05).

**Figure 6 pharmaceutics-15-00557-f006:**
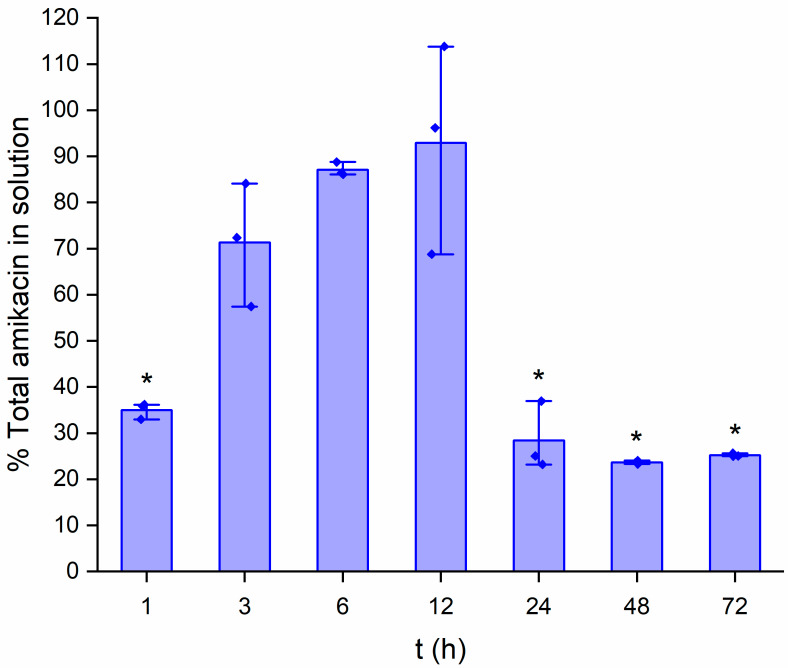
Amikacin release rate of a whole EPS film sample submerged in PBS (*n* = 3). Data were also evaluated for significance using a one-way repeated measures ANOVA and Tukey’s test post hoc. Versus point of maximum release (12 h): * *p* < 0.001.

**Figure 7 pharmaceutics-15-00557-f007:**
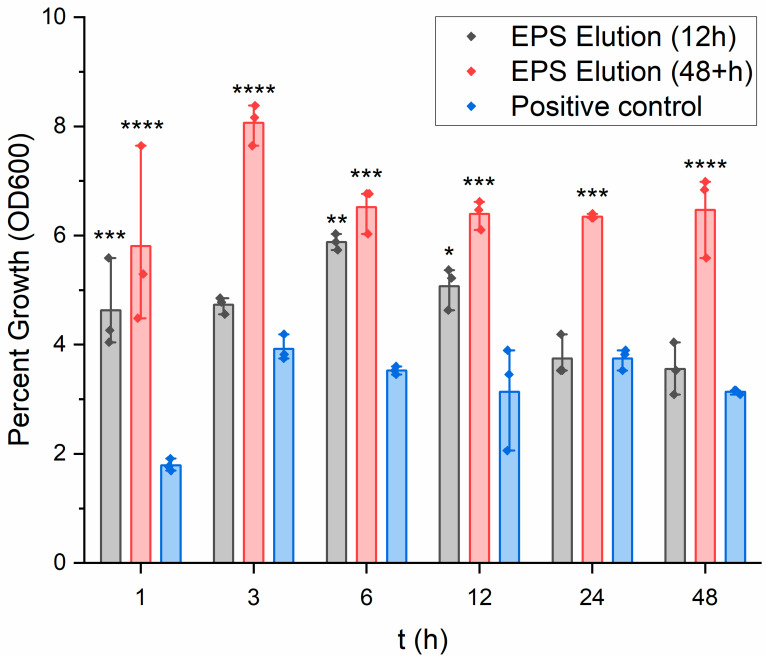
Comparing the effect on *S. aureus* growth inhibition using the EPS film elution taken at 12 h release, after 48+ h release, and using a positive control (high concentration of amikacin in water) (*n* = 3). Data were further evaluated for significance using a two-way repeated measures ANOVA and Tukey’s test post hoc. Versus positive control (high concentration of amikacin in water): * *p* < 0.01, ** *p* < 0.001, *** *p* < 0.0001, **** *p* < 0.000001.

**Table 1 pharmaceutics-15-00557-t001:** Variance in keratinocyte density among exposure groups. Values are the number of cells counted from random locations on the coverslips, presented as the mean followed by the standard deviation (*n* = 4).

	12 h	36 h
Control group (without EPSs)	112 ± 8.44	107 ± 6.68
Exposure group (with EPSs)	90.8 ± 5.89	94 ± 6.71

**Table 2 pharmaceutics-15-00557-t002:** Inhibition of *Staphylococcus aureus* data for eluted amikacin. Values are the mean of percent growth (OD600) subtracted from 100 to give inhibition percent, followed by standard deviation (*n* = 3).

Antibiotic	1 h	3 h	7 h	12 h	24 h	48 h
Amikacin (48+ h)	94.19 ± 1.34	91.93 ± 0.31	93.48 ± 0.35	93.60 ± 0.21	93.65 ± 0.03	93.53 ± 0.63
Amikacin (12 h)	95.37 ± 0.01	95.27 ± 0.0017	94.12 ± 0.0016	94.93 ± 0.0043	96.25 ± 0.0042	96.45 ± 0.0053
Positive Control	98.21 ± 0.09	96.08 ± 0.19	96.47 ± 0.06	96.86 ± 0.78	96.25 ± 0.16	96.86 ± 0.03

**Table 3 pharmaceutics-15-00557-t003:** Average mass of crude EPS produced per liter using two separate primary carbon sources. Values are the mean of 10 one liter EPS-producing cultures followed by standard deviation (*n* = 10).

	Dextrose	Corn Stover
Average crude EPS produced (grams)	1.24 ± 0.195	2.97 ± 0.452
Average grams EPSs per grams of substrate	0.207 ± 0.033	0.495 ± 0.075

**Table 4 pharmaceutics-15-00557-t004:** Recorded weights of films loaded with and without amikacin. The swelling ratio of the films was calculated using the average weight of the films not loaded with amikacin to represent the initial film weight.

Film Weights with Amikacin (g)	Film Weights with Amikacin (g, Average)	Film Weights without Amikacin (g)	Film Weights without Amikacin (g, Average)	Swelling Ratio (%)	Swelling Ratio (%, Average)
1.268, 1.281, 1.429, 1.302, 1.219	1.2998 ± 0.07	1.039, 0.955, 0.899, 0.976, 0.859	0.9456 ± 0.06	34.095, 35.470, 51.121, 37.690, 28.913	37.458 ± 7.42

## Data Availability

No Restrictions.

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
