# Peer review of "Thermophilic Exopolysaccharide Films: A Potential Device for Local Antibiotic Delivery"

_pharmaceutics, 2023, doi:10.3390/pharmaceutics15020557_

Round 1

Reviewer 1 Report

The manuscript describes the development of a new film made of thermophilic exopolysaccharides as a potential topical drug delivery tool. The work is innovative and exposes valuable and interesting clinical applications but, unfortunately, the assays made do not reflect all those potential applications. The physicochemical and cellular characterization of the film is rigorous; however, the microbiological characterization is poor. This contrast is an essential flaw of this study. Below, I suggest the possible reviews the author can make for improving the manuscript.

Major comments

The Introduction section is long. Although it contains important points, they are eclipsed by less important information. Please, shorten this section and make sure that the important ideas are clearer.

The Introduction section doesn’t highlight the importance of preventing infections, specially biofilm-related infections, a key type of infection that could be faced by using these films.

The manuscript lacks a Discussion section either independent or immediately after describing Results. Please, add some Discussion after your results.

In the Results section, I disagree with the results obtained from the release of amikacin over time. Aminoglycosides are susceptible to at least two kinds of chemical degradations in the experimental conditions the authors have used: the oxidation of the amino and hydroxyl functional groups (Graham AE, Speicher E, Williamson B. Analysis of gentamicin sulfate and a study of its degradation in a dextrose solution. J Pharm Biomed Anal. 1997;15:537–543.) and/or the cleavage under oxidizing conditions (Mullins ND, Deadman BJ, Moynihan HA, McCarthy FO, Lawrence SE, Thompson J. et al. The impact of storage conditions upon gentamicin coated antimicrobial implants. J Pharm Anal. 2016;6:374–381.). The crucial point is that these results coming from the amikacin degradation are also active against bacteria. Please, consider this information in your Discussion and redo the antibacterial experiments without basing that experiment on the release of amikacin.

The microbiological characterization is the main flaw of this study. Why did the author choose S. aureus and amikacin? There is a plethora of antibiotics that they could have chosen before this one: gentamicin or vancomycin, antibiotics used in the topical treatment of some biofilm-related infections. Aminoglycosides in monotherapy often induce the emergence of aminoglycoside resistance in staphylococci. However, amikacin would be the perfect antibiotic against another bacterium: Pseudomonas aeruginosa. Hence, I would offer two options to the author: changing the antibiotic and redoing the experiment with S. aureus or keeping amikacin but changing the bacterium from S. aureus to Pseudomonas aeruginosa.

The study lacks biofilm-related experiments. For instance: assays of biofilm development or treatment of 24-h biofilms. I suggest some works where the authors could inspire: assays of biofilm development (Antibiotics (Basel). 2021 Dec 3;10(12):1481. doi: 10.3390/antibiotics10121481.) and treatment of 24-h biofilms (https://www.frontiersin.org/articles/10.3389/fmicb.2019.02935/full).

Minor comments

The name Staphylococcus aureus or S. aureus is not properly written throughout the manuscript. This name is not sometimes in italics or “aureus” harbour a capital A. Please, correct these mistakes.

The word “versus” should be in italics. Please, correct this typo.

The word “Figure” is miswritten throughout the manuscript. Please, put this word with the first letter in capital.

In all Figures, please replace “Time (hours)” with “t (h)”.

All Figures contain a title that is repeated in the Figure captions. This information is redundant. Please, remove the title from all of them.

Reviewer 2 Report

The paper reports the development of thermophilic exopolysaccharide films, that are claimed “to be used for local antibiotic delivery.”

In my opinion, there are not enough studies on exopolysaccharide produced by Geobacillus sp. Strain WSUCF1 for local kanamycin delivery.

However, the manuscript is lacking supplementary experiments to claim that these films are good candidates for drug delivery systems. In fact, there is not enough data supporting the possibility to use the material as a biomaterial and potential applications are not foreseen. Moreover, the authors should re-structure the manuscript as it is in many parts confusing, being difficult to understand the results obtained. Also, the English should be fully revised. In general, the document presents interesting results but some of them should be merged or removed to improve readability. Sometimes, the manuscript simply describes a compilation of results that were obtained through several assays.

Thus, the authors need a major revision of the manuscript before publishing in Pharmaceutics Journal.

My comments are the following:

Abstract: 

-       This section is well written.

Keywords:

-       Geobacillus sp. could be added in the Keyword’s section.

Graphical abstract:

-       Adding a graphical abstract will be valuable for the manuscript.

Introduction:

-       The authors should be careful with the English and with the transition between sentences and paragraphs.

-       Add information about the importance of amikacin treat serious bacterial infections

-       Revise the last paragraph of the introduction section. From line 82 to line 89, the paragraph should be reformulated. The word “hypothesized” could be removed. In this part, the authors should summarize the main objective of the research work, and the techniques used, among others. It should be clear and straightforward.

Material and methods:

-       Line 113: the rotation of the centrifugation should be changed with g force instead of RPMs since the rotor size might differ, and g force will be different while rpm will stay the same.

-       Line 117: For extraction process, it always better to precipitate with cold ethanol at least twice.

-       Line 119: The authors referred that the EPS was obtained by centrifugation. It is important to explain how the polysaccharides are conserved. More information should be added, such as if the exopolysaccharide were in freeze-dried powder form?

-       Some details should be removed, like “the solution was then vortexed or mixed with a magnetic stir bar…”. It will be important to synthesize the description of the methods.

-       Uniformize the units: ml or mL. Units used in all the manuscript must be in accordance with the International System of Units.

-       Replace CO2 by CO2

-       Line 145: What is the resolution of the FTIR analysis?

-       Line 163: The authors should never begin a sentence with a numeral. Instead, you should try to reword the sentence or spell out the number.

-       Lines 194-202: This paragraph could be removed. This observation is not adequate for a scientific paper.

-       Line 207: some sentences seem to be incomplete such as “…cells adhere to.”

-       In general, the method’s section should be restructured. Some details are unnecessary, it seems like a logbook and not a research paper’s details. Some results/observations are described in this section.

-       Line 250: Amikacin was used at high concentration; it could be interesting to add information about the concentration.

Results and discussion:

-       The results are not always fully illustrated. Some figures could be added to better support the results.

-       The authors should be careful about the transitions between sentences such as “American hospitals. The antibiotic-loaded…” (Line 295-296). This statement should be verified in the manuscript.

-       The authors referred to flexibility (Line 300). This status/behavior should be evaluated by an adequate technique such as Rheometer or Dynamic mechanical analysis. The mechanical property a an important parameter to evaluate in the current research.

-       The FTIR analysis section should be better discussed. More regions presented in the spectra should be explained and discussed.

-       As the authors evaluated the 3 types of films by FTIR; it will be important to study also these 3 films by 1H-NMR. This spectroscopy technique is a powerful tool for obtaining complex carbohydrate’s structure and composition.

-       Line 332: “Both cell lines” which cell lines?

-       Line 339: The authors should be discussed the results obtained and compared them with the literature. It is important to highlight that any information that does not present the direct findings or outcome of the research should be left out of the results and discussion section.

-       Figures 3.a. and 3.b. Why the authors did not study the viability of keratinocytes exposed to EPS, as a control?

-       Figure 3.c. Why the authors study the viability of keratinocytes exposed to PHB film?

-       It is important to perform the experiments during the same frame time, some experiments were performed for 48 hours and others during 72h.

-       Figure 5. The authors should change the font of legend of the figure.

-       Line 451-455: This paragraph is not properly written.

-       Section 3.6. The discussion is too long. In this experiment, it will be interesting to study the drug delivery system during at least 28 days. It is important to highlight that “sustained-release dose forms are designed in such a way that the rate of drug release from the tablet matrix occurs in a controlled manner over an extended period of time maintaining a constant plasma drug level thus improving patient compliance and effective clinical outcomes.” https://www.ncbi.nlm.nih.gov/pmc/articles/PMC7227831/

-       Another important parameter is to evaluate the morphological/structural of these films developed.

-       The Table 2. was not cited in the text. The sentence “Values are the mean..” should be below the table. The same statement should be done for the other tables.

-       The section 3.8. In this section the authors were discussing the something different. Why this section? Dextrose vs corn stover. The referee is not understanding the objective of this experiment in the current manuscript.

-       The section 3.9. The discussion should be enhanced.

-       The section 3.10. It will be interesting to study the swelling during the time (for example 28 days)

-       Table 4. The results presented are very difficult to understand. Please reformulate the table.

-       There is no discussion of the results in the current paper. The authors should discuss the results obtained and compare them with those of other studies.

-       In general, some results presented are not necessary in the current manuscript. It will be important to reformulate the results and to better discuss the results obtained.

Conclusion:

This section should be revised and rewritten. It seems more like an introduction. Please state the most important outcome of your work: An overview of the main reason for the research, main findings and results, the scope for further research, and a strong concluding sentence should be discussed in the conclusion's section.

Reviewer 3 Report

Using turnitin I verified 81% similarity (a high percentage for work by the same author). I request clarification. The turnitin report is attached.The article is interesting. However, some relevant aspects must be considered. Below are listed some considerations

- Abstract: the justification for the relevance of the work must be revised. There are many natural polymers that are being used as carriers for drugs and/or other active compounds. The objective was not clearly defined.

-Introduction: the state-of-the-art use of exopolysaccharides as drug carriers and/or active compounds has not been evidenced. Searches in the area must be entered.

-Inform brand and models of all equipment used.

- Lines 128-132: confused. What is the purpose of this information. Were concentrations and methodologies based on any references?

-The dimensions of the samples must be presented. in all analyses, including controlled release tests, as they may affect the results. The mass is relevant, but so is the standardization of sample dimensions, as they affect the contact surface.

-2.6. MTT Assay: long and confusing text. Describe the methodology directly and objectively.

-In general, the wording of methodologies should be revised, written directly. Long texts with unnecessary information for the reproduction of the experiment.

-Statistical analysis: it is not clear the number of replicates and how the experiments were carried out. More details are needed.

-- Item "3.2. Uptake of EDC and Amikacin by FTIR": the main bands must be indicated in the spectra. The analyzes are only qualitative, it is not possible to discuss quantitative aspects without proper treatment of the data. Indications of functional groups in the spectra should be based on literature data.

-Were the viability tests not performed with a control film (without antibiotic)?

-The results obtained, in general, should be compared with other studies in the literature.

-Conclusions: long. It should be reviewed, considering the proposed objectives.

Round 2

Reviewer 1 Report

Though I am still missing biofilm-related experiments, I reckon that the manuscript has been improved and can be published.

Reviewer 2 Report

Manuscript has been corrected according to most of my suggestions and the required descriptions have been inserted into the text.